# Optimizing Exoskeleton Design with Evolutionary Computation: An Intensive Survey

Fabio Stroppa [1,*], Aleyna Soylemez [1], Huseyin Taner Yuksel [1], Baris Akbas [1] and Mine Sarac [2]

1   Department of Computer Engineering, Kadir Has University, 34083 İstanbul, Turkey;
    aleyna.soylemez@stu.khas.edu.tr (A.S.); huseyintaner.yuksel@stu.khas.edu.tr (H.T.Y.);
    akbassbars99@stu.khas.edu.tr (B.A.)
2   Department of Mechatronics Engineering, Kadir Has University, 34083 İstanbul, Turkey;
    mine.sarac@khas.edu.tr
*   Correspondence: fabio.stroppa@khas.edu.tr

**Abstract:** Exoskeleton devices are designed for applications such as rehabilitation, assistance, and haptics. Due to the nature of physical human–machine interaction, designing and operating these devices is quite challenging. Optimization methods lessen the severity of these challenges and help designers develop the device they need. In this paper, we present an extensive and systematic literature search on the optimization methods used for the mechanical design of exoskeletons. We completed the search in the IEEE, ACM, and MDPI databases between 2017 and 2023 using the keywords "exoskeleton", "design", and "optimization". We categorized our findings in terms of which limb (i.e., hand, wrist, arm, or leg) and application (assistive, rehabilitation, or haptic) the exoskeleton was designed for, the optimization metrics (force transmission, workspace, size, and adjustability/calibration), and the optimization method (categorized as evolutionary computation or non-evolutionary computation methods). We discuss our observations with respect to how the optimization methods have been implemented based on our findings. We conclude our paper with suggestions for future research.

**Keywords:** exoskeleton; design; optimization; evolutionary algorithms; mechanical design; robotics

## 1. Introduction

Exoskeletons can be observed in many terrestrial and marine organisms (such as bugs and shellfish) as a rigid, resistant component to provide protection, support, and sensation. Inspired by nature, wearable robotic devices can be designed to create exoskeletons for humans—either for their whole body [1] or for specific body limbs, such as arms [2], hands [3], legs [4], knees [5], wrists [6], etc.

Exoskeletons are used in many applications. When designed with no powered actuation, namely *passive* exoskeletons, they follow and/or track users' movements [7]. Tracked movements can be visually expressed in virtual environments [8] or used as the controller reference for a secondary robotic device during teleoperation tasks [9]. Alternatively, passive exoskeletons can distribute the weight of heavy objects on anatomical joints more equally to avoid potential injuries [10]. When equipped with batteries and actuators, namely *active* exoskeletons, they are used to augment and improve users' physical capabilities; examples range from performing heavy workload tasks [11] and completing sports activities [12] to assisting patients with physical/neurological disabilities while performing activities of daily living [13].

However, all these benefits come at a cost: exoskeletons are very challenging to be designed, implemented, and controlled [14]. Safety is the first and most important issue. Exoskeleton joints must align perfectly with anatomical joints to avoid potential harm, and their mechanical links and joints should work effectively within the workspace of human anatomical joints and natural degrees of freedom (DoFs). Depending on the

targeted application, exoskeletons must be adjustable for users of any size and ability level. The movements allowed by the exoskeleton should follow users' behavior naturally and without creating discomfort. Finally, these devices—especially assistive devices—should be as compact and lightweight as possible to enhance wearability. Thus, designs should include small actuator sizes, high output forces (in relation to the size of the actuator), and effective power transmission through the links. These issues and limitations can be overcome either by alternative design solutions (e.g., using softer rather than rigid mechanical links) or by exploiting optimization algorithms as tools for decision making. The integration between design and optimization techniques brings an open question: how do engineers decide which method or algorithm best optimizes different design criteria?

Mathematical optimization is the search for the best element within a set of alternatives based on specific criteria, and it is a common tool for solving engineering problems [15–17]. While the most conventional strategies focus on numerical and calculus-based methods [18], they might not be the best solution for engineering designs due to the properties of these problems: non-discrete domains, non-differentiability, multimodality, discontinuity, reliability, and robustness pose a challenge to classical methods. Alternative optimization methods exist to overcome these difficulties; among them, the nature-inspired methods of evolutionary computation (EC) [19,20] appear to be a common and effective way to deal with engineering optimization problems, and we argue that such methods are widely used in exoskeleton design and control.

This paper presents an extensive literature search on optimization studies for exoskeleton design. We compare the state of the art in designing exoskeletons based on optimization metrics, particularly focusing on the trend of evolutionary computation. As designers of exoskeletons (e.g., mechanical, mechatronics, or electronics engineers) come from different backgrounds, they might not know the latest trends or trade-offs in optimization methods. Our contribution is to investigate how exoskeletons are optimized, as well as how and why evolutionary computation is an effective solution in this field, and to provide effective suggestions for future research.

The rest of this work is organized as follows. In Section 2, we provide a background on exoskeletons and optimization methods. In Section 3, we report the findings of the literature search and provide a summary separating EC and non-EC methods. In Section 4, we discuss our findings and provide opinions. Finally, in Section 5, we conclude the paper.

*Search Strategy and Eligibility Criteria*

Figure 1 summarizes the search and elimination criteria. We completed the literature search in the MDPI, IEEE, and ACM databases from 2017 to 2023 (May) using the keywords "exoskeleton", "design", and "optimization". We investigated 722 studies from this search—155 from MDPI, 146 from IEEE, 187 from ACM, and 234 from Scopus. We eliminated studies for which the main focus was not the mechanical design of exoskeletons using optimization methods, along with 83 overlaps obtained from Scopus.

Specifically, we observed that 48 studies did not include any optimization method, 35 studies claimed to have implemented an optimization algorithm without giving further details, 57 studies focused on control and trajectory optimization rather than mechanical design, and 1 study used a trivial brute force algorithm rather than a proper optimization method [21]. Among the remaining studies, six optimized the mechanical design through finite element analysis (FEA) [22–27]. FEA is mostly embedded with current computer-aided design software to analyze how a given design reacts under real-world conditions rather than offering an exact solution that would correspond to a desired constraint or requirement. This resulted in 32 papers being included in the review.

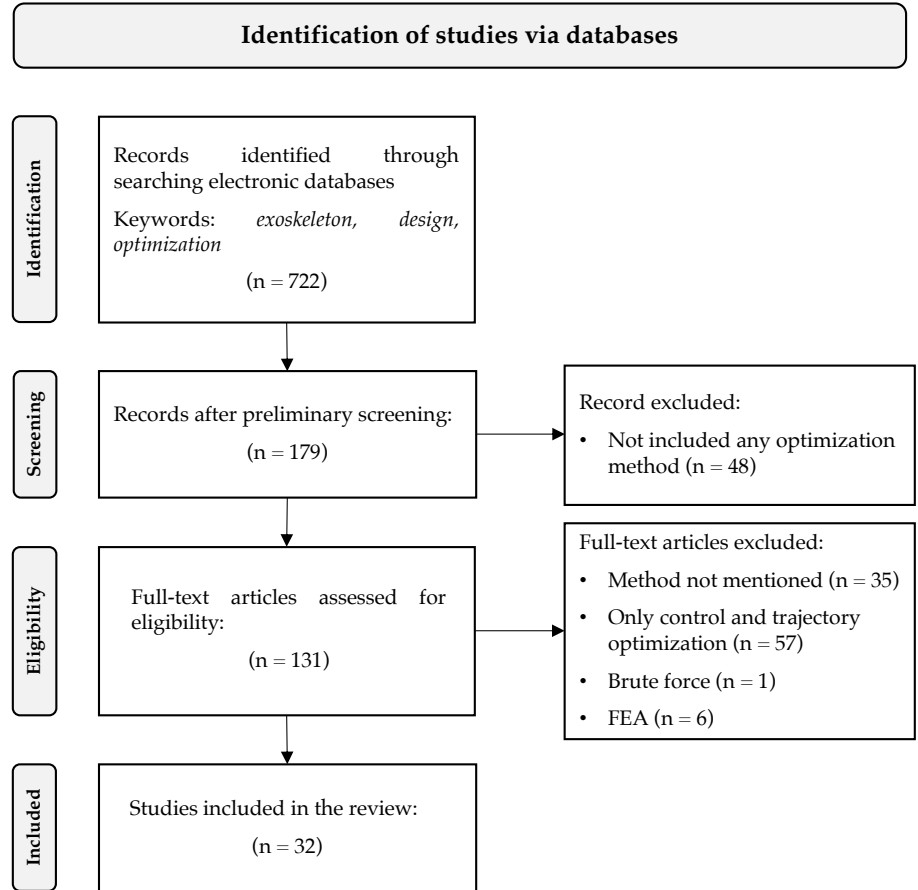

**Figure 1.** Flow diagram of the systematic literature review protocol.

## 2. Background

### 2.1. Background on Exoskeletons

Robotic exoskeletons are wearable devices with rigid links and mechanical joints corresponding to the anatomy of the human body. Even though different body locations have different anatomical properties and complexities that might require different design choices, the main variable defining an exoskeleton design is its application field. These application fields can be summarized as follows:

- **Assistance:** Assistive exoskeletons augment users' physical abilities to help them perform real-time activities that might be challenging to be completed alone. These devices can be used by (i) people with disabilities in their daily lives or (ii) healthy workers while performing physically demanding tasks in a workspace (see Figure 2a). Regardless of the target users, these devices must be capable of adapting their operation to perform different tasks or to interact with different objects. They must be portable, lightweight, and easy to wear while applying high interaction forces. They must achieve the range of motion of the anatomical joints without harming users when functionality limits are reached. Finally, these devices should feel highly transparent to follow the physical guidance of users. Observing/tracking movement performance is neither mandatory nor favorable.

- **Physical Rehabilitation:** Rehabilitative exoskeletons are used in clinical settings to treat patients suffering from physical or neurological disabilities (see Figure 2b). Due to users' limited functional capabilities, rehabilitative exoskeletons must be easy to wear without a predefined initial orientation (i.e., the device adapts to the patient's position rather than the opposite) and provide high output forces with respect to the actuator size adopted for the exoskeleton. The range of motion of anatomical joints must be achieved without harming patients when functionality limits are reached.

Patients should be able to perform different actions with no prior control or mechanical design change thanks to the devices' instant adaptability. Rehabilitation exoskeletons allow patients to actively participate in therapy exercises and monitor their progress in muscular activity. Unlike assistive devices, they are often grounded and do not need to be portable.

- **Haptic Rendering:** Haptic exoskeletons render an artificial sense of touch in response to virtual interactions or remotely operated real interactions (see Figure 2c). They must be wearable to track users' joint movements to control the interactions performed by virtual avatars or remote robots. Similarly to assistive devices, portability and instant adaptability are crucial. Haptic exoskeletons must feel highly transparent to follow users' physical guidance, especially when there is no interaction at the virtual/remote site. Since the target user profile is assumed to be healthy, the wearability or the amount of output forces is not as crucial as for other applications but is preferred.

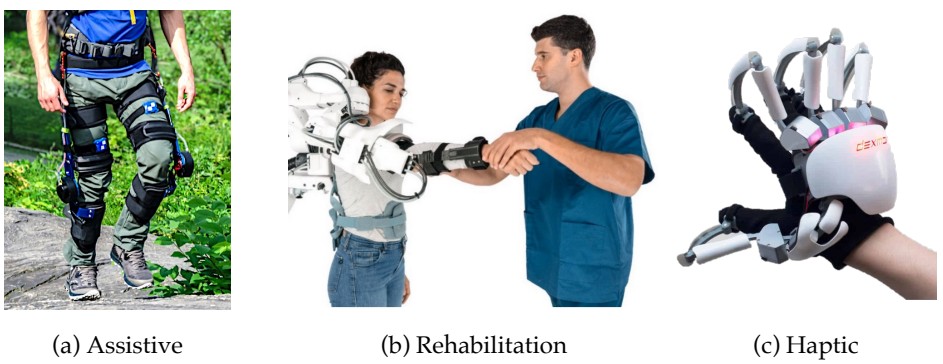

(a) Assistive               (b) Rehabilitation               (c) Haptic

**Figure 2.** Exoskeletons used in different applications [28–30].

The target applications require the exoskeletons to satisfy different design requirements. While some of these requirements should be achieved by design choices, such as suitable actuator technology, tracking strategy, or kinematic design, others should be achieved by optimizing the link lengths within a predefined kinematic chain. Table 1 depicts the mapping between these requirements and parameter/metrics to be optimized, which are detailed as follows:

- **Workspace:** The workspace is the range of motion the user is allowed while wearing an exoskeleton. Exoskeletons must respect the natural movements of users' limbs to ensure safety, and their mechanical limits must not exert force on human joints once they reach their natural limits. An exoskeleton must be comfortable, as users wear the device during operation. The kinematic and ergonomic design must be ensured not to cause any pain or fatigue.

  - While designing an exoskeleton, the mechanical joints must be aligned with the anatomical joints with minimal mechanical changes and cover the overall range of motion for the anatomical joints they are aligned with. This allows the exoskeleton to be inherently safe, ergonomic, and comfortable. To achieve this outcome, an optimization algorithm must retrieve the best link lengths and the actuated motion to either (i) *maximize* the operational workspace for each assisted joint or (ii) *maximize* a different design requirement and simultaneously ensure that the natural workspace is covered via constraints.

- **Force Transmission:** The human body has highly complex kinematics. For example, the human wrist can be modeled with three DoFs [31] (flexion, pronation, and radial deviation), the human finger with four DoFs [32] (one for the distal interphalangeal joint, one for the proximal interphalangeal joint, and two for the metacarpophalangeal joint), etc. The anatomic joints' complexity (and their proximity to each other) has led designers to decouple the actuators from the joints and transmit the actuator forces through linkage-based mechanical devices—whether they are made of rigid or soft

materials. In addition, linkage-based transmission allows designers to augment the transmitted forces through effective kinematic chains and lower the actuator size. The efficacy of its force transmission should be evaluated based on (i) the size of the wearable actuator components, (ii) the amount of force/torque rendered on the user's joints safely and comfortably, and (iii) the ratio between the actuated and output forces for each independent joint.

–   While designing an exoskeleton, an optimization algorithm should retrieve the best link lengths to *maximize* the force transmission for each assisted joint or for the overall targeted task (e.g., grasping a one-liter water bottle or lifting a five-kilogram storage box). Using such optimization techniques could also yield the same output forces with smaller actuators, improving the portability/wearability of the system as well.

- **Adjustability/Calibration:** Unlike prosthesis devices, exoskeletons are not custom-made for each potential user with different limb sizes. This lack of customization might cause misalignment, harm users, or work with a limited operational workspace or performance. In addition, especially for rehabilitative applications, wearing an exoskeleton should be equal and pain-free for every user.

    –   While designing an exoskeleton, an optimization algorithm should ensure the same performance for users of all sizes. There are three ways of achieving this outcome: (i) *maximize* the allowed range of limb sizes with no focus on other metrics, (ii) *maximize* the allowed range of limb sizes while optimizing another design requirement simultaneously, or (iii) *maximize* one of the previously detailed design requirements while ensuring an acceptable range of adjustability to different limb sizes via constraints.

- **Size:** While some full-arm exoskeletons need to be carried by a base due to their high weight [13], there is a great deal of research on reducing their weight and making them portable [33]. Exoskeletons can have improved portability by minimizing the mechanical components' size or weight.

    –   While designing an exoskeleton, an optimization algorithm should ensure the same performance with the smallest set of link lengths as much as possible by (i) *minimizing* the link lengths while other performance measures are fixed at a reasonable and predefined level via constraints or (ii) *maximizing* a different design requirement and simultaneously ensuring the acceptable set of link lengths to be covered via constraints.

**Table 1.** Mapping between design requirements and metrics to be optimized.

| Requirement | Metrics |
|---|---|
| User safety | Workspace, calibration |
| High output forces | Force transmission |
| Portability | Force transmission, size |
| Wearability | Calibration, size |
| Joint tracking | Calibration |
| Adaptability to different tasks | Workspace, force transmission |

The ultimate efficacy of an exoskeleton can be achieved by optimizing many features and factors simultaneously; therefore, choosing the best optimization technique is crucial for the ultimate performance of the device.

### 2.2. Background on Optimization and Evolutionary Computation

#### 2.2.1. Optimization Problems

Optimization is the mathematical process of searching for a set of *decision variables* ($x$) that would minimize or maximize one or more specific *objective functions* ($f_m$) while satisfying certain *constraints* (inequalities $g_j$, equalities $h_k$, and bounds) as expressed in (1).

While some problems focus on optimizing a single objective function, it is also common to encounter problems that require the optimization of more than one objective (namely, multi–objective optimization problems (MOOPs)). With MOOPs, there may be scenarios in which improving one objective might worsen the other(s), resulting in a conflict. In this case, rather than having a single optimal solution, we focus on obtaining a set of *trade–off* solutions that are *non-dominated* by any other solution of the problem, i.e., no other solution is better than those in all objectives. However, these trade–off solutions might be optimal for one objective but not for others [34,35]. This set of solutions is called the *Pareto* front.

$$
\begin{aligned}
\text{minimize/maximize} \quad & \{f_1(x), ..., f_M(x)\} \\
& x = \{x_1, ..., x_N\}^T \\
& x_i \in x, \quad i = 1, ..., N \\
\text{subject to} \quad & g_j(x) \geq 0, \quad j = 1, ..., J \\
& h_k(x) = 0, \quad k = 1, ..., K \\
& x_i^L \leq x_i \leq x_i^U
\end{aligned}
\tag{1}
$$

### 2.2.2. Optimization Methods

An algorithm or method of solving optimization problems provides a systematic and efficient way of creating and comparing new solutions to retrieve the optimal solutions. Optimization methods can be categorized as *exact* or *approximate* methods: the former retrieve the exact optimal solution and are usually based on direct or gradient-based methods [36], whereas the latter retrieve sub–optimal solutions that are acceptable approximations of the global optimal. For design problems, engineers usually rely on approximate methods because:

- Objective functions are usually complex (i.e., nonlinear, non-convex, discontinuous, discrete, mixed, and multimodal), and the effectiveness of the exact methods is not guaranteed [37];
- Robust solutions are preferred over global ones (i.e., when the optimal solution lies on a peak of the function, a small change in its value causes instability to the system, whereas a sub–optimal solution lying in a plateau area is less susceptible to change) [38];
- Reliable solutions are preferred over global ones, as uncertainty in the search space might lead to infeasible optimal solutions (i.e., in violation of the constraints) [39].

### 2.2.3. Evolutionary Computation

Evolutionary computation (EC) proposes some common approximate methods widely used in engineering. EC is a sub field of soft computing specialized in solving optimization problems [19]. Darwin's theory of natural selection inspires EC techniques; they generate a population of possible solutions to a provided problem and *evolve* toward the optimal solution. The insight on how to evolve toward the optimal solution (i.e., the *metaheuristic*) depends on the details of the EC algorithms, which may mimic natural selection [40], colonies of animals in search of food [41], or even cosmological creation [42]. Figure 3 shows a (partial) taxonomy of EC with its branches and different methods.

EC techniques are also population–based, since they propagate multiple solutions simultaneously rather than a single solution at a time. Thanks to this property, they can complete the search in parallel to cover a broader region of the search space, to prevent premature convergence, and to retrieve multiple optimal solutions if necessary (i.e., multi–modal or conflicting multi–objective problems) [43–45].

EC techniques are also preferred among engineers thanks to the simplifications in the problems' formalization. They usually do not require strict mathematical equations in their objective functions, which can be implicitly defined by the specific engineering problem to be solved (e.g., directly from simulations). Additionally, since they are independent of a mathematical model, the same algorithm can be used on different problems.

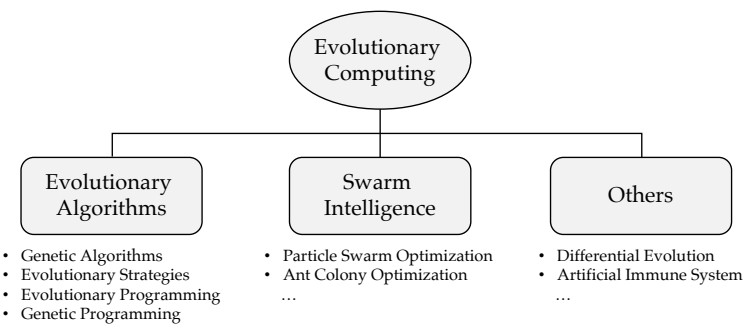

**Figure 3.** Partial taxonomy of evolutionary computation with its most common methods.

## 3. Mechanical Design Optimization

We have examined how exoskeleton design is optimized in the literature, mainly focusing on the different methods and techniques used to define and solve optimization. The advantages of EC motivated us to divide our findings into two categories: EC and non-EC techniques. In both categories, we investigated research studies in terms of which limb and application the exoskeleton was designed for, the optimization metrics, and the optimization method.

### 3.1. Exoskeleton Design with Evolutionary Computation Techniques

Table 2 reports the list of EC techniques used in the studies we evaluated, along with their abbreviation and a reference to the original article where the method was proposed, whereas Table 3 shows the studies in which EC techniques were used to optimize force transmission, workspace, weight, and size. We observed that the most commonly used EC techniques are genetic algorithms, particle swarm optimization, differential evolution, and evolutionary strategies.

**Table 2.** Evolutionary computation techniques in exoskeleton design.

| Abbreviation | Algorithm |
|:---:|:---:|
| GA | Single-Objective Genetic Algorithm [40,46] |
| NSGA–II | Elitist Non-Dominated Sorting Genetic Algorithm [47] |
| WBGA | Weight-Based Genetic Algorithm [48] |
| PSO | Particle Swarm Optimization [49] |
| DE | Differential Evolution [50] |

**Table 3.** Exoskeletons optimized with evolutionary techniques in terms of application (assistance (A), rehabilitation (R), or haptic (H)), optimization metric (force transmission (FT), workspace (W), or size (S)), and optimization method.

| Authors | Limb | Application | Metrics | Optimization Method |
|:---:|:---:|:---:|:---:|:---:|
| Li et al. [51] | Hand | A | FT, W | NSGA–II [52] |
| Du J. et al. [53] | Hand | R | FT | GA [40,46] |
| Lee et al. [54] | Wrist | R, H | FT, W | NSGA–II [47] |
| Hunt et al. [55] | Arm | A | FT | NSGA–II [47] |
| Tschiersky et al. [56] | Arm | A | FT | GA [40,46] |
| Du Z. et al. [57] | Arm | A | FT, W | PSO [49] |
| Zakaryan et al. [58] | Arm | R | FT, S | DE [50,59] |
| Yoon et al. [60] | Arm | A | W | WBGA [47] |
| Asker et al. [61] | Leg | A | FT, W | WBGA [48] |
| Deboer et al. [62] | Leg | A | FT, S | NSGA–II [47] |
| Paez et al. [63] | Leg | A | FT, W | NSGA–II [47] |
| Tian et al. [64] | Leg | A | S | PSO [49] |
| McDaid [65] | Leg | R | W | WBGA [48] |
| Xu et al. [66] | Leg | R | FT | PSO [49,67] |
| Rituraj et al. [68] | Leg | A, R | S, FT | NSGA–II [47] |
| Yu et al. [69] | Leg | A | FT | WBGA [48] |

3.1.1. Genetic Algorithms (GAs)

Evolutionary Algorithms (EAs), in particular genetic algorithms (GAs), are the most popular branch of EC techniques, as they directly implement the process of natural selection and survival of the fittest [40,46]. They (i) generate a population of random solutions within the search space of the problem (namely *individuals* or, to resemble genetic processes, *chromosomes*), (ii) assign a *fitness* value to each solution by evaluating the objective function based on their values, and (iii) generate new solutions by mixing the values of individuals in the current population (a process named *crossover*). By allowing only the most fitting individuals to perform crossover and to be preserved in the next generations, GAs evolve their population, converging to the optimal solution of the problem. The crossover operation exploits the features of good solutions to speed up convergence to an optimum; however, there is no guarantee for the optimum to be global rather than local. Therefore, GAs implement an additional operator inspired by genetic *mutation*, which randomly modifies values of a newly generated solution to favor exploration of the search space and escape local optima.

Besides the advantages of every population-based method mentioned in Section 2.2.3, GAs are easy to implement and very efficient in converging to (sub)optimal solutions. On the other hand, one disadvantage is that their many genetic operators come with many parameters, and fine-tuning these parameters might be non-trivial and primarily based on trial and error. Furthermore, since GAs are iterative stochastic methods, they might be inefficient in solving the optimization problem in real time or when the target limb is simulated to reach different poses consecutively. The MATLAB Optimization Toolbox offers an implementation of GA with the command ga.

In our literature review, we found the following research studies that used GAs to optimize the mechanical design parameters of exoskeletons over a **single objective function**:

- Du J. et al. [53] retrieved the optimal link lengths, maximizing the force transmission of their hand exoskeleton for rehabilitation; and
- Ttchiersky et al. [56] retrieved the optimal actuator placement, maximizing the force transmission of their shoulder exoskeleton for assistive use.

The rest of the research studies using GAs optimized the mechanical design parameters of exoskeletons over **multiple objective functions**. There are several GAs specifically developed to approach MOOPs, which we will detail separately.

Weight-Based Genetic Algorithm (WBGA)

WBGA is based on the simplest classical method of solving an MOOP: scalarizing a set of objectives into a single objective by multiplying each objective by a user-supplied weight, namely the weighted sum method [35]. Setting up appropriate weights for each objective might be challenging, as their importance depends on the context of the problem (note that all weights should add up to 1). To ensure that the Pareto front is uniformly and fully explored, one should solve the same MOOP with different sets of weights. This repetitive process is fastened with EC techniques thanks to their population-based nature: each individual is assigned a different set of weights. In doing so, the whole population maintains multiple weight vectors, finding multiple Pareto-optimal solutions in a single run instead of one Pareto-optimal solution corresponding to a particular weight set.

Since WBGA transforms a problem from multi- to single-objective, it is computationally inexpensive with respect to other multi–objective EAs (the evaluation runs in $\mathcal{O}(MN)$, where $M$ is the number of objectives, and $N$ is the number of individuals in the population). Unfortunately, simplicity and computational efficiency come with some disadvantages: (i) each objective function needs to be normalized, (ii) it is difficult to handle mixed optimization problems (i.e., when some objectives are of the maximization type and others are of the minimization type), (iii) a uniformly distributed set of weights is not guaranteed to generate a well-distributed set of solutions on the Pareto front, and (iv) this method

cannot generate solutions for non-convex Pareto fronts. The latter is a significant drawback in engineering problems, as objective functions usually come from iterative scripts rather than analytical formulas, so the shape of the Pareto front is not guaranteed to be convex.

In our literature review, we found the following research studies that used WBGA to optimize the mechanical design parameters of exoskeletons over multiple objective functions:

- Yoon et al. [60] retrieved the optimal joint locations that minimize (i) the misalignment of the exoskeleton and (ii) the frame protrusion of their shoulder exoskeleton for assistive use;
- Asker et al. [61] retrieved the optimal link lengths that (i) maximize the force transmission and (ii) minimize the misalignment between a human and their knee exoskeleton for assistive use;
- McDaid [65] retrieved the optimal link lengths that (i) maximize the workspace, (ii) maximize the distance from robot singularities, and (iii) minimize the size of their leg exoskeleton for rehabilitation; and
- Yu et al. [69] retrieved the initial conditions that minimize (i) displacement and (ii) dynamics of the hydraulic cylinder composing the joint of their knee exoskeleton for assistive use.

Elitist Non-Dominated Sorting Genetic Algorithm (NSGA–II)

Elitism is a strategy in EC where the best individuals—the elites—are retained in the population over different generations to speed up convergence. In single-objective optimization, elites are individuals with better objective function values, whereas in multi–objective optimization, an individual is considered elite depending on its proximity to the Pareto front (i.e., all non-dominated solutions as defined in Section 2.2.1 are elites). Therefore, algorithms such as NSGA–II [47] rank individuals in the population based on which non-dominated set they belong to. NSGA–II evaluates the dominance of each individual by collecting the non-dominated individuals in a set and repeats the process without non-dominated individuals until every individual in the population belongs to a specific and progressively ranked set. Due to elitism, individuals in lower ranks are eliminated, which allows the algorithm to converge to the Pareto front. Additionally, to provide diversity in the population and fully explore the Pareto front, NSGA–II uses a niching operator (the crowding distance [47]) that penalizes individuals in a crowded area of the objective or decision-variable space.

Elitism has the advantage of not allowing non-dominant individuals to be removed from the population, which helps convergence. However, this also has the disadvantage of losing some non-dominated individuals when the whole population is composed of only elites, which mostly happens in late generations. Like other multi–objective EAs, the time complexity of the method is expensive (the evaluation runs in $\mathcal{O}(MN^2)$, where $M$ is the number of objectives, and $N$ is the number of individuals in the population—although in practice, sorting is performed on a population of size $2N$). A variant of NGSA-II is implemented in the MATLAB Optimization Toolbox with the command `gamultiobj`.

In our literature review, we found the following research studies that used NSGA–II to optimize the mechanical design parameters of exoskeletons over multiple objective functions:

- Li et al. [51] retrieved the optimal link lengths that (i) maximize the force transmission and (ii) minimize the difference between contact forces and reduce the ejection phenomenon (i.e., fingers push the targeted object away instead of grasping it) while satisfying an accepted range of motion on their hand exoskeleton for assistive use;
- Lee et al. [54] retrieved the optimal rotational angles and joint distribution that maximize (i) the accuracy and dexterity through an index estimating the kinematic performance of specific posture (the global condition index) and (ii) the minimum distance between the links and the centerline (the interference safety margin) of their wrist exoskeleton for rehabilitation and haptic use;

- Hunt et al. [55] retrieved the optimal actuator location, maximizing its stiffness volume both for (i) translation and (ii) rotation of their shoulder exoskeleton for assistive use;
- Deboer et al. [62] retrieved the optimal link lengths, spring stiffness values, angles, and displacements that minimize (i) the peak power and (ii) the total length of the two actuators of their leg exoskeleton for assistive use;
- Paez et al. [63] retrieved the optimal link lengths and joint locations that minimize (i) the moment load at the joint, (ii) the difference between the natural human posture and that observed while wearing the exoskeleton to promote a natural torso motion, and (iii) the torque deviation to match a linear profile while independently maximizing the torque output of their knee exoskeleton for assistive use; and
- Rituraj et al. [68] retrieved the optimal link lengths and angles that minimize (i) the maximum distance between the actuators and (ii) the load on the device of their assistive/rehabilitative knee exoskeleton for rehabilitation and assistive use.

### 3.1.2. Swarm Intelligence (SI)

Swarm intelligence (SI) is an EC technique based on the collective behavior of self-organized systems. These systems are composed of agents locally interacting with each other and their environment, as inspired by swarms, herds, or flocks of animals gathering to locate food or build colonies [70]. Although different methods are classified as SI, the most used method in engineering is particle swarm optimization (PSO) [49]. Similarly to GAs, PSO generates a population of individuals evolving toward the optimal solution. Just like birds in a flock change their direction based on the choreography of the individual who locates food, each individual in PSO learns from their own and other members' experience and adapts their behavior by changing the search pattern. This iterative and cooperative process makes the population—in this case, the *swarm*—converge to the optimal solution.

PSO is computationally inexpensive in time and space compared to EAs, exhibiting better performance, a higher convergence rate, and higher-quality solutions than GA [71,72]. PSO also has different multi–objective optimization versions [67], with the main disadvantage of poorly controlling diversity in the swarm of solutions converging to the Pareto front.

In our literature review, we found the following research studies that used PSO to optimize the mechanical design parameters of exoskeletons:

- Du Z. et al. [57] independently retrieved (i) the optimal link lengths that minimize the misalignment between human and robot and (ii) the optimal rope position of the cable-driven mechanism that maximizes the torque actuating their arm-support exoskeleton for assistive use;
- Tian et al. [64] retrieved the optimal link lengths that minimize the force transmission of their leg exoskeleton for wheelchair assistive use to support the user from sitting to standing; and
- Xu et al. [66] used a multi–objective PSO [67] to retrieve the optimal design parameters of a magnetorheological actuator that maximize (i) the force transmission and (ii) workspace of their leg exoskeleton for rehabilitation.

### 3.1.3. Differential Evolution (DE)

Differential evolution (DE) is designed to optimize problems over continuous domains by representing individuals as vectors and using vector differences to perturb the population [50]. A further difference relative to EAs is that mutation is performed on the distribution of individuals in the current population such that search directions depend on the location of the individuals selected to calculate mutation values. Similarly to PSO, the main disadvantages of DE come when dealing with MOOPs. Although its convergence rate to the Pareto front seems to be high, it does not reach the actual Pareto front but, rather, a nearby (sub–optimal) and non-diverse front.

In our literature review, we found the following research studies that used DE to optimize the mechanical design parameters of exoskeletons:

- Zakaryan et al. [58] used a weight-sum-based DE [59] to retrieve the optimal link and joint weights that minimize (i) the total mass of the device, (ii) the maximal magnitudes of cable tensions, and (iii) the maximal difference between magnitudes of agonist–antagonist cable tensions (i.e., to resemble the structure of the natural muscular system of human limbs) of their arm exoskeleton for rehabilitation.

### 3.2. Designs with Other Optimization Techniques (Non-Evolutionary)

Table 4 reports the list of non-EC techniques used in the studies we evaluated, along with their abbreviation and a reference to the original article where the method was proposed, whereas Table 5 shows the studies in which non-EC techniques were used to optimize force transmission, workspace, compliance, weight, and size. We observed that the most commonly used non-EC techniques are the interior point algorithm, the Levenberg–Marquardt algorithm, the Simplex algorithm, Pareto local search, the goal attainment method, and geometric differentiation.

**Table 4.** Non-evolutionary computation techniques in exoskeleton design.

| Abbreviation | Algorithm |
|---|---|
| LMA | Levenberg–Marquardt Algorithm [73,74] |
| GD | Geometric differentiation [75] |
| IPA | Interior point algorithm [76,77] |
| GAM | Goal attainment method [78] |
| PLS | Pareto local search [79] |
| SA | Simulated annealing [80] |
| NMSM | Nelder–Mead simplex method [81] |

**Table 5.** Exoskeletons optimized with non-evolutionary techniques in terms of application (assistance (A), rehabilitation (R), or haptic (H)), optimization metric (force transmission (FT), workspace (W), size (S), or adjustability/calibration (AC)), and optimization method.

| Authors | Limb | Application | Metrics | Optimization Method |
|---|---|---|---|---|
| **Amirpour et al. [82]** | Hand | H | W, AC | LMA [73,74] |
| **Bianchi et al. [83]** | Hand | R | FT, S | LMA [73,74] |
| **Liang et al. [84]** | Hand | A, R | W | GD [75] |
| **Xu et al. [85]** | Hand | R | W | IPA [76,77] |
| **Qin et al. [86]** | Hand | R | AC | GAM [78] |
| **Secciani et al. [87]** | Hand | A | AC | IPA [76,77] |
| **Kulkarni et al. [88]** | Wrist | A | W | IPA [76,77] |
| **Vatsal et al. [89]** | Arm | A | FT | PLS [79] |
| **Balser et al. [90]** | Arm | A | FT | IPA [76,77] |
| **Vazzoler et al. [91,92]** | Arm | A | FT | IPA [76,77] |
| **Anderson et al. [93]** | Leg | A, R | FT | IPA [76,77] |
| **Malizia et al. [94]** | Leg | A | W | SA [80], NMSM [81] |
| **Xiao et al. [95]** | Leg | A | FT | IPA [76,77] |
| **Kim et al. [96]** | Leg | A | AC | IPA [76,77] |
| **Bougrinat et al. [97]** | Ankle | A | S | IPA [76,77] |

### 3.2.1. Interior Point Algorithm (IPA)

Interior point algorithms (IPAs) approach linear and non-linear convex optimization problems [76]. Given an objective function, they iteratively retrieve the optimal solution by traversing the interior of the feasible search space (i.e., within the problem's constraints). This class of algorithms can solve linear programming problems in polynomial time [77]. However, their gradient-based nature requires the objective function to be differentiable (even though implementations for non-differentiable functions can also be found [98]). The MATLAB Optimization Toolbox offers an implementation of IPA with the command `fmincon`.

In our literature review, we found the following research studies that used IPA to optimize the mechanical design parameters of exoskeletons:

- Xu et al. [85] retrieved the optimal link lengths that minimize the difference between the workspace covered by the human and their hand exoskeleton for rehabilitation;

- Secciani et al. [87] retrieved the optimal geometrical parameters that allow the kinematics to minimize the error to the desired trajectories of their hand exoskeleton for assistive use;
- Kulkarni et al. [88] retrieved the optimal link lengths that minimize the difference between the workspace covered by the human and their wrist exoskeleton for assistive use over two joints (one joint is optimized, while the other joint is modeled as an inequality constraint);
- Balser et al. [90] retrieved the optimal parameters of the cable-driven actuator (i.e., radius, cable, and pulley) and link lengths that minimize the difference between the torques generated by the human and their shoulder exoskeleton for assistive use;
- Vazzoler et al. [91,92] retrieved the optimal joint spring parameters and positions that minimize the root mean square between joint torques of their leg exoskeleton for assistive use to distribute the weight effectively;
- Anderson et al. [93] retrieved the optimal "mechanical parameters" (The authors left the description of these mechanical parameters intentionally abstract, and they reported that "the system-specific mechanical design will determine which variables affect the model outputs") that minimize the reflected inertia of their leg exoskeleton for rehabilitation and assistive use;
- Kim et al. [96] retrieved the optimal position and pressure of pneumatic actuators that minimize the average energy consumption rate of the human joint while running using their leg exoskeleton for assistive use;
- Xiao et al. [95] retrieved the optimal link lengths and structural angle that minimize the torque exerted by their knee exoskeleton for assistive use; and
- Bougrinat et al. [97] retrieved the optimal link length and angles that maximize the artificial lever arm (i.e., the perpendicular distance between the joint and the line of action) of their ankle exoskeleton for assistive use.

### 3.2.2. Levenberg–Marquardt Algorithm (LMA)

The Levenberg–Marquardt algorithm (LMA), also known as the damped least-squares method, is an iterative procedure based on gradient descent specifically used to solve nonlinear least-squares problems (i.e., curve fitting) [73,74]. Due to its gradient-based nature, LMA requires the objective function to be differentiable.

In our literature review, we found the following research studies that used LMA to optimize the mechanical design parameters of exoskeletons:

- Amirpour et al. [82] retrieved the optimal link lengths that (i) minimize the difference between worst-case workspace dexterity and isotropy and (ii) minimize the distance between the angle of the forces exerted on finger phalanges and the perpendicular direction on their hand exoskeleton for haptic use; and
- Bianchi et al. [83] retrieved the optimal link lengths that minimize the maximum value of the torque of their hand exoskeleton for rehabilitative use.

### 3.2.3. Geometric Differentiation (GD)

Geometric differentiation (GD) is a branch of geometric calculus (including geometric algebra, differentiation, and integration) that can be used as an optimization method for engineering design problems when the desired device must follow a specific trajectory [75].

In our literature review, we found an interesting case in which designers proposed their own optimization method based on geometric differentiation:

- Liang et al. [84] retrieved the optimal joint shape (i.e., link lengths and their poses) that minimizes the joint workspace and trajectory of humans and the hand exoskeleton for rehabilitation and assistive use. They particularly focused on designing a flexible joint based on the anatomy of grasshoppers (a model that can also be observed in crustaceans such as crabs and lobsters). They proposed a GD-based optimization algorithm (*Congjugate Surface Optimization Algorithm*) that evaluates the fingertip's path during flexion and extension motion.

### 3.2.4. Goal Attainment Method (GAM)

The goal attainment method (GAM) is a weighted-sum algorithm for multi–objective optimization based on sequential quadratic programming [78]. Its specific formulation turns a multi–objective problem into a single-objective problem in which the weights allow the objectives to be under- or overachieved, enabling the designer to be relatively imprecise about the initial design goals. Despite being based on a weighted sum, this method can discover solutions even if the Pareto front is non-convex. The MATLAB Optimization Toolbox offers an implementation of GAM with the command `fgoalattain`.

In our literature review, we found the following research studies that used GAM to optimize the mechanical design parameters of exoskeletons:

- Qin et al. [86] retrieved the optimal link lengths that minimize the misalignments between the finger joints (one objective for each joint, for a total of two joints) and their hand exoskeleton for rehabilitation.

### 3.2.5. Pareto Local Search (PLS)

Pareto local search (PLS) is a method used to solve MOOPs based on ND trees, which is a specific data structure implemented to store solutions based on their dominance ranking [99]. PLS is very effective when applied to solve biobjective combinatorial optimization problems, but it becomes inefficient for more than two objectives [79].

In our literature review, we found the following research studies that used PLS to optimize the mechanical design parameters of exoskeletons:

- Vatsal et al. [89] retrieved the optimal design parameters (joint angles, spring parameters, moments, and forces) that minimize muscle effort rates during realistic dynamic tasks (each muscle is defined as a separate objective function, resulting in an MOOP) while using their shoulder exoskeleton for assistive use.

### 3.2.6. Nelder–Mead Simplex Method (NMSM)

The simplex algorithm is an iterative numerical process to find the optimal solution to a linear programming problem by converting objectives and constraints into a system of linear equations [100]. The Nelder–Mead variant can solve non-differentiable problems or problems with discontinuities in the search space [81]. Its main advantage is its independence from the gradient information and its significant improvement of solutions in the first few generations. However, it is time-consuming and may fail to converge to the global optimum when near a local optimum. The MATLAB Optimization Toolbox offers an implementation of NSMS with the command `fminsearch`.

In our literature review, we found the following research studies that used NMSM to optimize the mechanical design parameters of exoskeletons:

- Malizia et al. [94] retrieved the optimal properties (spring location, spring resting length, and angle between spring and leg) that minimize the error between the desired torque and the actual torque exerted by their leg exoskeleton for assistive use.

### 3.2.7. Simulated Annealing (SA)

Simulated annealing (SA) is a stochastic local search method for optimization [80]. It mimics the heat treatment used in metallurgy that alters a material's physical and chemical properties to increase its ductility and reduce its hardness (e.g., the process used to manufacture sword blades). SA generates a random point in the search space and transposes it following the layout of the objective function: if the next point has a better value, then SA accepts it and reiterates the procedure; if not, SA accepts it with a probability depending on how much worse the point is with respect to the previously generated point and how long the algorithm has been running to simulate *temperature* (i.e., hot at the beginning and cold toward the end). The probability is calculated based on the Boltzmann distribution. SA has the advantage of escaping local optima; however,

the cooling simulation must be very slow to enforce regularities of the objective's layout, resulting in long runs.

In our literature review, we found the following research studies that used SA to optimize the mechanical design parameters of exoskeletons:

- Malizia et al. [94] enhanced the results of the primary optimization method (as discussed in Section 3.2.6, the authors primarily used NMSM) by outdistancing the starting parameters of each optimization attempt from the optimal values found in the previous attempt on their leg exoskeleton for assistive use.

## 4. Discussion

While conducting an extensive literature survey on research studies implementing optimization algorithms for the mechanical design of exoskeletons, we observed some trends, issues, and challenges. In this section, we first detail our observations from the perspective of how optimization methods are implemented and presented, then provide recommendations for future designers to improve their research and contributions to the research community.

### 4.1. Discussion on Optimization Methods

#### 4.1.1. Wrong use of the term "MOGA"

While conducting this literature review, we observed that there is a misuse of the term MOGA, as it can be used as an acronym for any multi–objective genetic algorithm. However, the term MOGA indicates the specific GA for solving MOOPs proposed by Fonseca and Fleming [52] in which less-dominated solutions have higher fitness than highly-dominated solutions and the MOOP becomes the maximization of a single objective. In our survey, we found that Li et al. [51] used the term MOGA to identify a generic multioptimization GA, but the algorithm they used is actually NSGA–II, not MOGA.

Therefore, we usually use the term MOEA, which stands for multi-optimization evolutionary algorithm, to indicate a generic multi–objective genetic algorithm and avoid possible misinterpretations.

#### 4.1.2. Popularity of NSGA–II and Interior Point Algorithm

We observed that NSGA–II is the most common EA method to solve engineering MOOPs. We hypothesize that this is thanks to being computationally faster than other MOEAs and allowing for higher diversity in the Pareto front. Another reason for being very popular is that it is embedded in the MATLAB Optimization Toolbox with the command `gamultiobj`, so designers do not need to have complete knowledge of the algorithm specifics and can use it as a black box, likely making it the fastest MOEA in the literature. Even so, we found that most of the studies reviewed in this survey specifically mentioned the name NSGA–II, showing that the authors have sufficient knowledge about EA rather than simply relying on a black-box method. There are a few exceptions to this observation: Hunt et al. [55] only reported that the MATLAB Optimization Toolbox was used, without mentioning NSGA–II, whereas Rituraj et al. [68] stated that they used NSGA–II in the MATLAB environment, although it is not clear if they used the toolbox or recreated the algorithm from scratch. Lastly, Paez et al. [63] reported that NSGA–II allowed them to mutate over the possible populations of solutions through a definition of the objectives, which clearly refers to the ability to select the appropriate design parameters out of a pool of trade-off solutions rather than specifying fixed preferences among objectives and obtaining only one solution (e.g., WBGA).

Similarly, we observed that IPA is the most common non-EA method for designing exoskeletons. The reason is likely the same: it is implemented with the `fmincon` command of the MATLAB Optimization Toolbox and can be used for constrained, non-linear, multivariable functions.

### 4.1.3. Simplified Implementation of WBGA

Our survey revealed that WBGA is also commonly used when approaching engineering MOOPs. As discussed in Section 3.1.1, the reason is that the simplest way to address a MOOP is by transforming it into a single-objective optimization problem through the weighted sum method. Any multi–objective optimization algorithm exploiting this technique maintains its original computational complexity without the burden of sorting non-dominated solutions. This way, an exoskeleton designer does not need to know MOOP or MOEA theory and can still solve their design problems by guaranteeing solutions on the entire Pareto front if it is convex.

However, according to our observations, most researchers do not implement a full version of WBGA, which works with different sets of weights to explore the entire Pareto front. Instead, they simply assign prefixed weights to each objective function based on their preference—i.e., they numerically score each objective, quantifying the priority of one against another using only a single set of weights. Since a single set of weights generates a single solution on the Pareto front, there is no opportunity to compare it with the other solutions on the front.

Another point usually missed by researchers is how to prepare the objective functions and the weights for the optimization process. First, the sum of all weights must be 1 (or 100 in cases in which the weights are expressed as percentages). Secondly, the objective functions must be normalized to have comparable ranges of magnitudes to avoid objectives overshadowing each other in the sum. We observed that these preparations were ignored in most of the studies reported in this survey—or at least it was not clearly mentioned by the authors.

Although WBGA, a preference-based method, may be helpful in cases in which designers have a clear idea of which objective requires higher priority, in practice, quantifying the weights is not straightforward, especially when dealing with more than two objective functions. A more exacerbated case is when one objective must be strictly optimized and others are less important: giving any weight value to the less critical objectives will take away from the main objective. For example, when designing a kinematic chain that can reach one or more desired points, we want to solve the inverse kinematics (i.e., minimize the distance between the end effector and the desired point) but also minimize the chain's link lengths to save production cost. In such a case, we are not interested in any solution that does not reach the desired point, and saving cost can be considered as less crucial.

While investigating the studies included in this survey, we observed different ways of setting the weights for WBGA. Asker et al. [61] set the weights as a normalization factor to assign each term of the objective function the same weight when searching for the optimal value. Similarly, Yu et al. [69] used weights of one, assigning their objectives the same values (i.e., in terms of classic weighted-sum method, both objectives have a weight value of 0.5). Assigning each objective the same weight means that we are not interested in obtaining global optimal values in any of the conflicting objectives but have equally mediocre values, as shown in Figure 4a. This might be desired in some cases, but we argue that the authors did not make this choice consciously, since they did not report any justification for their preference among objectives. We can assume most of the studies included in this survey applied a weighted-sum method to their GA, unaware of the formal algorithm of WBGA that allows solutions to explore the front.

The correct implementation of WBGA [48] each individual in the population (or groups of them) a different set of weights, usually uniformly distributed such that the (convex) Pareto front is fully explored (as detailed in Section 3.1.1). McDaid [65] presented the only implementation of WBGA with more than a single set of weights. However, the sum of each set was not unified, and the sets were not uniformly distributed along the Pareto front; instead, they set them with an order of magnitude more significant over each other. The author claimed that this provided more flexibility than ideal multi–objective approaches, such as non-dominated sorting, as in NSGA–II, but without providing a direct comparison

between the two methods. We believe that the word they used to justify the use of WBGA, "flexibility", is too vague and needs systematic comparisons with further details.

Finally, we debate that using WBGA might not be the best approach for design optimization problems. Weighted-sum methods cannot retrieve solutions in a non-convex region of the Pareto front (see Figure 4b), and the design problems associated with exoskeletons have non-linear objective functions obtained from scripts or simulations, where the convexity of the Pareto front is not guaranteed. In summary, most authors have used WBGA because it is the easiest MOEA to understand, implement, and execute. However, we observed that many authors used this method incorrectly due to a lack of knowledge of MOOP/MOEA theory. We would like to highlight the importance of correctly implementing methods and invite future designers to pay more attention to these details, especially to set the weights between different objectives.

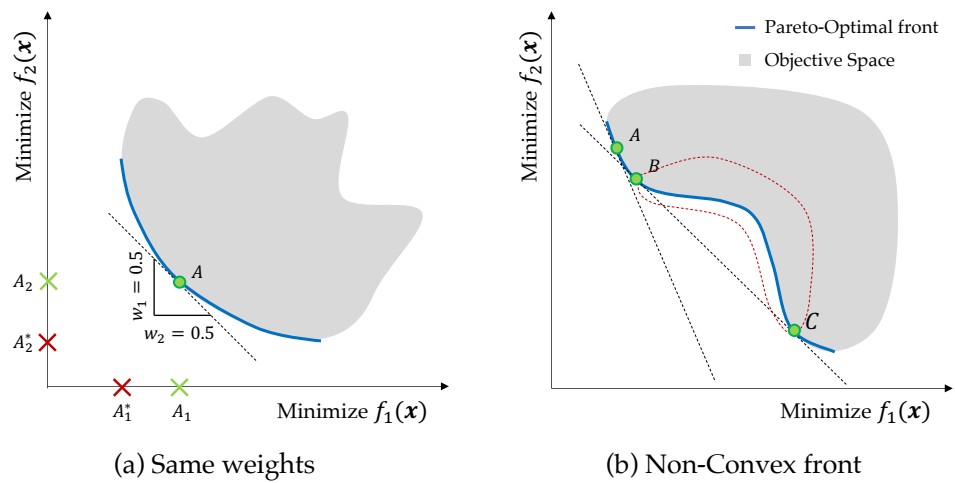

(a) Same weights                  (b) Non-Convex front

**Figure 4.** Examples of solutions retrieved with weighted-sum methods for a generic bi–objective minimization problem. Each set of weights represents a different Pareto-optimal solution defined by the slope of the line intersecting the front. In (**a**), by assigning the same value to each weight, only the trade–off solution ($A$) is retrieved; better values for each objective were ignored (e.g., $A_1^*$ is the minimum of $f_1$, whereas $A_2^*$ is the minimum of $f_2$, and $A$ is exactly in between). In (**b**), no solution can be found between $B$ and $C$ (the front inscribed in the red hull) because other solutions in the convex section can be found with the same slope, which would better optimize the objective with a larger weight (i.e., with higher priority).

### 4.1.4. When to Use Different EC Methods

With sixteen studies relying on EC and thirteen relying on non-EC techniques, our survey revealed that EC is popular when designing exoskeletons. For example, McDaid [65] motivated their decision to use a GA by reporting that EAs are well-suited to nonlinear problems, as many optimization techniques can be employed. While this is a valid reason, several other aspects make EC suitable in engineering, with different algorithms that feature specific properties to be exploited.

### Multi Modality

Multi modality is one of the properties that can easily be addressed using EC techniques. Problems featuring many local optima raise conflict among accuracy, reliability, and computation time, and traditional optimization methods may fail to provide efficient results. For example, Vazzoler et al. [91,92] reported to have restarted the running of a non-EC technique (IPA) several times due to the method's inability to escape local optima.

On the other hand, EC techniques are population-based, which allows them to retrieve multiple solutions in a single run without the need to restart the method. However, evolving a population of solutions is still insufficient to avoid being trapped in local optima,

especially considering that the crossover operator of EAs promotes convergence in the region defined by the affected individual (which could be in the range of attraction of a local optimum). While other non-EC methods need several restarts in different points of the search space (such as IPA or other non-population-based methods), EAs can use mutation to escape local optima and ensure diversity in the population. In accordance with this point, Yoon et al. [60] relied on a GA to solve their design problem featuring multiple local optima; however, they did not specify the probability of mutation they used nor how exactly they addressed the problem, which could have been helpful to other designers facing the same issue.

There might be some cases in which the problem requires retrieving all optimal solutions (both local and global). Then, there are additional methodologies that can be employed: the crowding model [101], the sharing function [102], specific strategies such as adding a further objective that minimizes the derivative of the first objective (i.e., such that if the derivative reaches zero, the solution is a local optimum) and retrieves weakly dominated solutions with NSGA–II [44], or innovative algorithms such as the strengthening evolution-based differential evolution with prediction strategy [103].

Multi Objective

When it comes to solving MOOPs, EAs are still more efficient than classical methods due to their population-based nature. However, they also become computationally expensive: (i) their run time grows linearly with the number of objectives, as all of them must be evaluated; and (ii) every solution in the population needs to be compared against each other for non-dominated sorting. This is likely why most of the studies examined in this survey optimized only two objectives in their design. There are a few exceptions featuring more than two objectives, and we observed some issues regarding how they were implemented or reported. McDaid [65] and Zakaryan et al. [58] solved three-objective problems with weighted-sum-based EAs, which do not require non-dominated sorting but have the drawbacks described in Section 4.1.3. Paez et al. [63] solved a four-objective problem with NSGA–II; however, the last one was optimized independently, and the authors did not specify whether a single-objective GA was appositely used for that objective or how it was included in the whole problem without conflicting with the other three.

An interesting case was reported by Kulkarni et al. [88], who used a non-EC method (IPA) to solve a biobjective problem. However, since the IPA is a method for single-objective optimization, the authors modeled the second objective as an equality constraint. We argue that this is an inadequate way to address MOOPs for two main reasons. First, when aiming at a specific value (e.g., zero), we do not consider any other value feasible, no matter how close it comes to the desired value. This makes the algorithm lose its heuristic for convergence; although this depends on the way constraints are handled [104], this is mostly the case. Secondly, equality constraints are notoriously difficult to satisfy due to the floating-point precision representing the data (i.e., achieving a value exactly equal to zero is unlikely). The conventional method to handle them is to transform them into two inequality constraints by introducing a certain threshold, which increases the computational burden of the problem.

Choosing the Right MOEA

Although the state of the art presents numerous MOEAs, the most famous ones appear to be NSGA–II [47]—as confirmed by our survey—and SPEA–2 [105], which has not been used for exoskeleton design according to our extensive literature search. SPEA–2 seems to provide better solutions in terms of convergence and diversity but at the expense of computational time [106]. However, a clear ultimate winner between NSGA–II and SPEA2 cannot be determined [107], so both methods can be considered equivalent.

For an engineering problem such as designing exoskeletons, the objective functions are expected to have high computational complexity (e.g., often resulting from scripts or simulations). Therefore, the complexity of the optimization algorithm might play a

decisive role. Assuming that the number of objectives is smaller than the number of individuals in the population (thus negligible), NSGA–II runs in $\mathcal{O}(N^2)$, whereas SPEA–2 runs in $\mathcal{O}(N^2 \log \sqrt{N})$, where $N$ is the number of individuals. NSGA also presents a newer, improved version, namely NSGA–III [108], which can solve problems with up to fifteen objective functions with the same time complexity as its predecessor.

Hybrid Techniques: Combining Different Optimization Methods

Another interesting trend in solving engineering design problems is to combine different optimization methods, namely *hybrid techniques*. MOEAs can reach the region near the Pareto front relatively quickly, but it can take considerable time to achieve convergence. Therefore, we can use an MOEA for a small number of generations to approach the Pareto front, then use those solutions as an initial point for another optimization solver that is faster and more efficient for a local search—for example, the Goal Attainment method described in Section 3.2.4 as directly proposed by the MATLAB Optimization Toolbox [43]. In our survey, we found some examples of hybrid techniques, although they did not use EC. For example, Malizia et al. [94] used SA to set the starting parameters and NMSM for the overall optimization, whereas Vatsal et al. [89] attempted to implement convex optimization (using an unmentioned method) but became trapped in a local minimum, which they used as the initial condition for the Pareto Local search optimizer.

*4.2. Recommendations for Future Engineer Designers*

4.2.1. Advantages of EC over Other Methods for Mechanical Design of Exoskeletons

While conducting this survey study, we observed that the literature has a similar amount of work for EC and non-EC methods. We acknowledge that the choice of optimization method depends on various factors, including the specific problem at hand, available computational resources, and the expertise of the designer. In addition, non-EC methods (or classical optimization methods) might be favorable and effective in specific situations, particularly when the design space is well-defined and the objectives are simple and easily differentiable. However, in light of the findings reported in this survey, as well as the solid theory they are built upon, we would like to emphasize that EC offers powerful approaches for optimizing the mechanical designs of exoskeletons. We can summarize their advantages as follows.

Non-Differentiable and Complex Design Spaces

Exoskeleton design spaces often involve non-linear relationships and complex interactions between multiple design variables. Classical optimization methods such as gradient-based techniques rely on the assumption of differentiability, which may not hold in such scenarios. In contrast, EC methods can handle non-differentiable and non-linear design spaces more effectively.

Global Optimization in Wide or Multi–Objective Search Spaces

Exoskeleton design optimization usually involves finding the best combination of design variables to achieve multiple objectives, such as maximizing strength, minimizing weight, and ensuring user comfort. Classical methods often struggle to find the global optimum in multi–objective and highly non-linear problems. EC excels at global optimization by exploring a diverse range of solutions concurrently and maintaining a population of potential solutions.

Solution Exploration and Diversity

EC techniques employ mechanisms like mutation, recombination, and selection to explore the design space effectively. This exploration allows the algorithms to discover novel and potentially better solutions that may be missed by classical optimization methods. Additionally, EC methods maintain diversity within the population, preventing premature convergence to sub–optimal solutions and enabling the discovery of a wider range of trade–offs.

Robustness to Noisy or Incomplete Data

Exoskeleton design optimization often involves dealing with uncertain or noisy data, such as sensor measurements, material properties, or human biomechanical factors. Classical methods may be sensitive to noise and struggle to find reliable solutions in such scenarios. In contrast, EC methods can handle noisy or incomplete data more robustly. They can adapt and converge to good solutions even when the available information is limited or subject to uncertainty.

Encouraging Innovative Designs

EC allows for unconventional and innovative solutions to be explored. By introducing random variations and exploring a diverse set of solutions, EC methods can discover novel designs that may not be immediately obvious to human designers or classical optimization methods. This capability can lead to breakthroughs in exoskeleton design by discovering new and efficient solutions that were previously unexplored.

### 4.2.2. Importance of Using Mathematical Optimization in Engineering

While conducting our literature search, we eliminated 48 research papers out of 179 that passed our first screening process for not implementing an optimization method (see Figure 1) and one case implementing a brute force algorithm [21]. Researchers commonly include workspace or force-distribution optimization as their future work after designing and manufacturing an exoskeleton rather than employing optimization methods as part of the first development process. This is particularly interesting because the presented results from preliminary works can be subjected to profound improvements once optimization methods are implemented. Even though not all design-related activities are expected to be completed in a single research article, future researchers should still be more mindful of using at least simple, straightforward, and MATLAB-integrated optimization techniques to claim meaningful and comparative performance findings.

### 4.2.3. Considerations for Optimal Control

Optimal control aims to "determine the control signals that will cause a process to satisfy the physical constraints and simultaneously minimize some performance criterion" [109]. While the motivation of optimal control seems to lie in the same direction as mathematical optimization for mechanical design, the methods and techniques are significantly different. To focus on a specific targeted area, we decided to limit our search to studies optimizing the mechanical design parameters, eliminating 57 out of 131 research studies assessed for eligibility. Nevertheless, we would like to emphasize the popularity of optimal control techniques on exoskeletons and invite future researchers to present an independent systematic review with this specific focus.

### 4.2.4. Considerations for Structural Optimization

Out of the 38 research studies focusing on the mechanical optimization of exoskeletons, we eliminated 6 using FEA as the optimization method [22–27]. FEA is a computerized method for predicting how a product reacts to real-world forces, vibration, heat, fluid flow, and other physical effects, showing whether a product will break, wear out, or work the way it was designed. FEA analyzes how a given design reacts under real-world conditions rather than offering an exact solution corresponding to a desired constraint or requirement. While we also acknowledge the importance of optimizing the durability of mechanical links against actuator forces and human dynamics, these research papers still lack the mechanical properties (such as joint locations, actuator properties, link lengths, etc.) that would result in optimal operation metrics (such as force transmission over user limbs, allowed workspace, etc.). However, we report the interesting case of Jafarpour et al. [110], who combined FEA with MOEAs to study the mechanical response of insects' exoskeletons to normal contacts—not really related to robots, but on exoskeletons in nature.

### 4.2.5. Importance of Detailing the Optimization Method

Working on this extensive literature search, we focused on detailing the parameters and techniques used by each study as much as possible. One of the biggest issues with the current state of the art is the lack of information about optimization methods and parameters, as most of the authors only reported to have optimized their device, with no further specifics provided. This issue led us to discard 35 research studies from our literature search—more than the number of studies we eventually included (e.g., Xu et al. [111], Hua et al. [112], and Nguyen et al. [113]).

Ultimately, we presented the findings of 32 research studies systematically. While most of these studies clearly reported their findings and their motivations for their specific choices, we also observed a lot of studies that were published despite being somewhat vague. To present a complete survey, we made an extra effort to investigate their references. For example, Xu et al. [66] reported that "*the multi–objective algorithm was employed*" without directly specifying which algorithm they used. However, based on their reference to Gudmundsson et al. [114], we realized that they might have used either a single-objective optimization EA (specifically (1 + 1)ES [115]) or a multi–objective PSO [67]. Similarly, Yu et al. [69] did not specifically mention which optimization method they employed; by tracking down their references, we realized that they used the reliability design optimization proposed by Wang et al. [116]. In this paper, the authors specifically mentioned that they used a GA to retrieve the Pareto optimal solutions for their MOOP and combined the objectives using the weighted-sum method; however, as mentioned in Section 4.1.3, their weights were both equal to one, creating potential problems.

While we agree that these authors vaguely indicated their decisions during their research process, they should have chosen clear and informative ways of communicating research-based activities with future researchers. In many studies, the optimization problem is not well-defined, and readers might have difficulty understanding what the decision variables are, how the objective function is defined, and whether the objective must be minimized or maximized. We strongly believe that as researchers, our purpose is to share knowledge with future engineers as much as to develop new technologies.

## 5. Conclusions

In this paper, we presented an extensive and systematic literature search on the optimization methods used for the mechanical design of exoskeletons. We have presented our findings in terms of which limb (i.e., hand, wrist, arm, or leg) and application (assistive, rehabilitation, or haptic) the exoskeleton was designed for, the optimization metrics (force transmission, workspace, size, and adjustability), and the optimization method (categorized as evolutionary computation and non-evolutionary computation methods). We also prepared a detailed discussion with our observations on the implementation of optimization methods and recommendations for future designers.

According to our observations, evolutionary computation methods are commonly used in exoskeleton design. This is an expected outcome, thanks to their many properties such as updating several solutions simultaneously (population-based), their independence from gradient information, their fast convergence, and their ability to handle mixed optimization problems (including multimodal and multiobjectives).

The biggest limitation of our survey is the clarity and the detail provided by the authors of the research studies we included in our search. We would like to highlight this challenge and invite future authors to promote detailed and informative writing in their works in a clear and concise fashion.

**Author Contributions:** Investigation, A.S., H.T.Y. and B.A.; data curation, A.S., H.T.Y. and B.A.; visualization, A.S., H.T.Y. and B.A.; writing—original draft preparation, F.S.; writing—review and editing, F.S. and M.S.; supervision, M.S. All authors have read and agreed to the published version of the manuscript.

**Funding:** This research obtained no external funding.

**Institutional Review Board Statement:** Not applicable.

**Informed Consent Statement:** Not applicable.

**Data Availability Statement:** Not applicable.

**Conflicts of Interest:** The authors declare that they have no conflicts of interest regarding the publication of this paper.

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
