# Peer review of "Optimizing Exoskeleton Design with Evolutionary Computation: An Intensive Survey"

_robotics, doi:10.3390/robotics12040106_

Round 1

Reviewer 1 Report

In this article, the author conducted a systematic literature survey on the optimization method for exoskeleton mechanical design, and then analyzed and introduced it from three aspects: the auxiliary part of the exoskeleton, the application of the exoskeleton, and the optimization method of the mechanical structure of the exoskeleton. Finally, the optimization method of the exoskeleton is discussed, and suggestions are made for future engineers is related fields. Through the review, the author observed that evolutionary computing is more commonly used in exoskeleton design than non-evolutionary computing. However, deep analysis is limited in the paper, and the innovation and value is not obvious. Please find the comments as follows:

1.     The construction of an exoskeleton system includes three parts: structural design, establishment of human-machine coupling model, and controller design. As the author described in section 4.2.2, the article only organizes and analyzes the optimization method of mechanical design parameters, and does not consider the optimization method of control strategies in the exoskeleton system. This may limit the innovation of the article, resulting in limited content expressed in the article.

2.     In Section 3, various optimization methods of evolutionary computing and non-evolutionary computing and their advantages and disadvantages are described in detail. However, the article does not introduce in detail the advantages of EC compared with non-EC in the optimization of exoskeleton mechanical structure parameters. 

3.  It would be better if the author can provide more examples in exoskeleton design with EC.   

4. Is there a reference citation error in section 4.1.4 (line 693) of the article?

Moderate editing of English language is suggested.

Author Response

Reviewer #1

In this article, the author conducted a systematic literature survey on the optimization method for exoskeleton mechanical design, and then analyzed and introduced it from three aspects: the auxiliary part of the exoskeleton, the application of the exoskeleton, and the optimization method of the mechanical structure of the exoskeleton. Finally, the optimization method of the exoskeleton is discussed, and suggestions are made for future engineers is related fields. Through the review, the author observed that evolutionary computing is more commonly used in exoskeleton design than non-evolutionary computing. However, deep analysis is limited in the paper, and the innovation and value is not obvious. Please find the comments as follows:

Q1.1) The construction of an exoskeleton system includes three parts: structural design, establishment of human-machine coupling model, and controller design. As the author described in section 4.2.2, the article only organizes and analyzes the optimization method of mechanical design parameters, and does not consider the optimization method of control strategies in the exoskeleton system. This may limit the innovation of the article, resulting in limited content expressed in the article.

Thank you for this comment. We understand how this might come as a limitation with the amount of detail provided in the first submission. We have initially investigated the research studies focusing both on mechanical design, optimal control, and trajectory design with specific algorithms. While categorizing the optimization methods, we have realized that there are very few intersections between using the optimization methods for mechanical and control aspects of exoskeletons. In addition, we realized that focusing on every aspect would have conflicted with keeping the manuscript in a reasonable (and digestible) number of pages; thus, we decided to only focus on the mechanical design rather than reduce discussions and conceptual details to a minimum. Based on these considerations, we believe this decision allowed us to deliver a more specific and detailed aspect of design, and we do not consider it a limitation.

Q1.2) In Section 3, various optimization methods of evolutionary computing and non-evolutionary computing and their advantages and disadvantages are described in detail. However, the article does not introduce in detail the advantages of EC compared with non-EC in the optimization of exoskeleton mechanical structure parameters. 

Thank you for this suggestion. As we wrote the article, we considered the whole text as a discussion on why EC might be more beneficial than classical methods for exoskeleton designs. However, we agree with the reviewer that this should be clearly and directly stated in the text in a well-defined section – such that our article does not leave anything to the readers' interpretation. We do not think that Section 3 was the right place for such a discussion, as Section 3 is purely dedicated to reporting what we found in the literature; therefore, we have added a new Section 4.2.1 in the discussion (and the former Sections 4.2.1 to 4.2.4 have been shifted):

“4.2.1. Advantages of EC over Other Methods for Mechanical Design of Exoskeletons

While conducting this survey study, we observed that the literature has a similar amount of work for EC and non-EC methods. We acknowledge that the choice of optimization method depends on various factors, including the specific problem at hand, available computational resources, and the expertise of the designer. In addition, non-EC methods (or classical optimization methods) might be favorable and effective in specific situations, particularly when the design space is well-defined, and the objectives are simple and easily differentiable. However, in light of the findings reported in this survey, as well as the solid theory they are built upon, we would like to emphasize that EC offers powerful approaches for optimizing the mechanical designs of exoskeletons. We can summarize their advantages as follows.

Non-differentiable and Complex Design Spaces

Exoskeleton design spaces often involve non-linear relationships and complex interactions between multiple design variables. Classical optimization methods, such as gradient-based techniques, rely on the assumption of differentiability, which may not hold in such scenarios. In contrast, EC methods can handle non-differentiable and non-linear design spaces more effectively.

Global Optimization in Wide or Multi-Objective Search Spaces

Exoskeleton design optimization usually involves finding the best combination of design variables to achieve multiple objectives, such as maximizing strength, minimizing weight, and ensuring user comfort. Classical methods often struggle with finding the global optimum in multi-objective and highly nonlinear problems. EC excels at global optimization by exploring a diverse range of solutions concurrently and maintaining a population of potential solutions.

Solution Exploration and Diversity

EC techniques employ mechanisms like mutation, recombination, and selection to explore the design space effectively. This exploration allows the algorithms to discover novel and potentially better solutions that may be missed by classical optimization methods. Additionally, EC methods maintain diversity within the population, preventing premature convergence to suboptimal solutions and enabling the discovery of a wider range of trade-offs.

Robustness to Noisy or Incomplete Data

Exoskeleton design optimization often involves dealing with uncertain or noisy data, such as sensor measurements, material properties, or human biomechanical factors. Classical methods may be sensitive to noise and struggle to find reliable solutions in such scenarios. In contrast, EC methods can handle noisy or incomplete data more robustly. They can adapt and converge to good solutions even when the available information is limited or subject to uncertainty.

Encouraging Innovative Designs

EC allows for unconventional and innovative solutions to be explored. By introducing random variations and exploring a diverse set of solutions, EC methods can discover novel designs that may not be immediately obvious to human designers or classical optimization methods. This capability can lead to breakthroughs in exoskeleton design by discovering new and efficient solutions that were previously unexplored.”

Q1.3) It would be better if the author can provide more examples in exoskeleton design with EC.   

Thank you for this suggestion. We have conducted our literature search to cover studies in the past 5.5 years – considering that 2023 was also included in the search criteria until the date of the first submission of the manuscript, which was May – in databases like IEEE, ACM, MDPI, and (in the current review) Scopus. A span of 5 years is a standard range of time to investigate the recent trends in research, and the main limitation to the number of examples in exoskeleton design in EC is due to the number of existing studies in that range. We would also like to highlight that the latest search on Scopus did not produce any results on EC methods besides the ones we already had included in the first version of the manuscript. We hope that with the proposed work, we can highlight the advantages of EC methods over the non-EC methods for future designers to increase their use in exoskeleton engineering.

Q1.4) Is there a reference citation error in section 4.1.4 (line 693) of the article?

Thank you for noticing this error and pointing it out. We fixed the reference in the new version of the manuscript.

Reviewer 2 Report

The paper presents an extensive and systematic literature search on the optimization methods used for the mechanical design of exoskeletons.

The manuscript is well written, the content is presented in an interesting manner and the reading and the reading flows well.

A positive aspect of the work is that the analysis is applied to all exoskeletons, regardless of their field of use and the body district they vicariate.

That said, I have only a few minor observations:

- Why is Scopus not among the databases evaluated? I suggest the authors add it to give more robustness to the collection of reviewed articles.

- The date on which the search was conducted is missing. The years to which the articles refer are indicated, but since 2023 is only partially covered the date when the databases were searched should be indicated.

- How many of the articles selected in the various databases overlapped?

- L 104 is indicated that exoskeletons for rehabilitation should provide high output forces. This on the one hand depends on the body district involved in the rehabilitation (e.g., it is true for an exoskeleton for gait rehabilitation, but not for an exoskeleton for hand rehabilitation), but even assistive exoskeletons require high output forces or torques. 

I am not a native English speaker, but in my opinion the English used is appropriate and as mentioned above the reading is smooth.

Author Response

Reviewer #2

The paper presents an extensive and systematic literature search on the optimization methods used for the mechanical design of exoskeletons. The manuscript is well written, the content is presented in an interesting manner and the reading and the reading flows well. A positive aspect of the work is that the analysis is applied to all exoskeletons, regardless of their field of use and the body district they vicariate. That said, I have only a few minor observations:

Q2.1) Why is Scopus not among the databases evaluated? I suggest the authors add it to give more robustness to the collection of reviewed articles.

Thank you for this suggestion. In the updated manuscript, we have also added the findings of the studies found on Scopus. Particularly, our keywords gave us 234 papers from 2017 to 2023. We removed 83 of them as they were overlapping with papers from IEEE and MDPI, leaving 151 documents.

Out of these, only 11 were related to mechanical design and not related to control or trajectory optimization. However, we found the following:

  • Four papers were not related to exoskeletons, although the keywords should have removed non-exoskeleton designs from the search;
  • One paper was on optimal control, so we removed it from our search;
  • One paper was related to insects’ exoskeletons – even though it is not related to our purpose, we mentioned it in the manuscript since we believed it to be an interesting case:

“We report the interesting case of Jafarpour et al. [1], which combined FEA with MOEAs to study the mechanical response of insects' exoskeletons to normal contacts – not really related to robots, but on exoskeletons in nature.”

  • One paper was eliminated because the optimization method was not specified;
  • One paper used brute force, and we briefly mentioned it in Sec. 4.2.1 [2].

Ultimately, we were left with three new articles, which we added to the manuscript as follows:

“Bianchi et al. [3] retrieved the optimal link lengths that minimize the maximum value of the torque of their hand exoskeleton for rehabilitative use;

 Bougrinat et al. [4] retrieved the optimal link length and angles that maximize the artificial lever arm (i.e., the perpendicular distance between the joint and the line of action) of their ankle exoskeleton for assistive use; and

 Secciani et al. [5] retrieved the optimal geometrical parameters that allow the kinematics to minimize the error to the desired trajectories of their hand exoskeleton for assistive use.”

We would like to report that, to our surprise, none of these studies were relying on EC techniques. Two [4,5] used the Interior-Point algorithm, whereas the last one [3] used the Levenberg–Marquardt Algorithm.

Q2.2) The date on which the search was conducted is missing. The years to which the articles refer are indicated, but since 2023 is only partially covered the date when the databases were searched should be indicated.

Thank you for this comment. We have updated our search until the date of the first draft submission (May 2023). We added this detail in the text.

Q2.3) How many of the articles selected in the various databases overlapped?

Thank you for this comment. We have focused on databases from main publishers, such as IEEE, ACM, and MDPI. Therefore, we originally did not observe any overlap. When we extended our search on Scopus, as you suggested, our keywords gave us 234 papers from 2017 to 2023. We removed 83 of them as they were from IEEE and MDPI and therefore overlapped with previous findings.

Q2.4) L 104 is indicated that exoskeletons for rehabilitation should provide high output forces. This on the one hand depends on the body district involved in the rehabilitation (e.g., it is true for an exoskeleton for gait rehabilitation, but not for an exoskeleton for hand rehabilitation), but even assistive exoskeletons require high output forces or torques. 

Thank you for this comment. We understand how our wording might have created confusion in this manner. The hand exoskeleton's output forces should be much lower than the gait exoskeletons. However, what was intended in the original text is to focus on “high output forces compared to the actuator/transmission size allowed for each body size.” At this point, we never intended to compare the exoskeletons for different body locations but rather for other applications. A higher “force to actuator size” ratio is required for assistive devices to help participants perform more challenging grasping or operation tasks – whether the assistance is given to the hand or the leg. Similarly, a higher “force to actuator size” ratio is required for rehabilitative devices to overcome the level of spasticity on patients’ limbs – regardless of which limb. However, haptic devices might be sufficiently successful even with a lower “force to actuator size” ratio since the main motivation is creating noticeable and believable touch during virtual or remote interactions. We have adjusted our wording as high output forces with respect to the actuator size adopted for the exoskeleton in multiple places in the manuscript.

Reviewer 3 Report

The authors' intensive study on optimization methods used for the mechanical design of exoskeletons is interesting and useful for future engineer designers. Although the identification of studies through databases narrowed the scope of the used publications to 29, the authors review over 100 literature sources and analyze the results in detail with respect various limbs, applications ,optimization indicators and optimization methods.

The detailed discussion of the authors' observations on application of optimization methods as well as the recommendations for future designers is a good base for future research in this area.

On line 693 is given a character ? when referring to the literature.

Author Response

Reviewer #3

The authors' intensive study on optimization methods used for the mechanical design of exoskeletons is interesting and useful for future engineer designers. Although the identification of studies through databases narrowed the scope of the used publications to 29, the authors review over 100 literature sources and analyze the results in detail with respect various limbs, applications, optimization indicators and optimization methods.

The detailed discussion of the authors' observations on application of optimization methods as well as the recommendations for future designers is a good base for future research in this area.

Q3.1) On line 693 is given a character? when referring to the literature.

Thank you for noticing this error and pointing it out. We fixed the reference in the new version of the manuscript.

Round 2

Reviewer 1 Report

The revised manuscript is improved a lot. 

Moderate editing of English language is required.